# Multiplexed Digital PCR Reference Gene Measurement for Genomic and Cell-Free DNA Analysis

**DOI:** 10.3390/cells14191544

**Published:** 2025-10-03

**Authors:** Dilek Yener, Eloise J. Busby, Jo Vandesompele, Gertjan Wils, Susan D. Richman, Henry M. Wood, Jim F. Huggett, Carole A. Foy, Alison S. Devonshire

**Affiliations:** 1National Measurement Laboratory, LGC, Guildford GU2 7XY, UKjim.huggett@lgcgroup.com (J.F.H.);; 2Leeds Institute of Medical Research at St James’s, University of Leeds, Leeds LS9 7TF, UK; 3Pxlence, Building RTP (Entrance 97), UZ Gent Campus, Corneel Heymanslaan 10, 9000 Ghent, Belgium; jo.vandesompele@pxlence.com (J.V.); gertjan.wils@pxlence.com (G.W.); 4NIHR Leeds Biomedical Research Centre, Leeds LS7 4SA, UK; 5Faculty of Health & Medical Sciences, University of Surrey, Guildford GU2 7XH, UK

**Keywords:** precision medicine, dPCR, reference gene stability, genome equivalents, liquid biopsy

## Abstract

**Highlights:**

**What are the main findings?**
A five-gene multiplex digital PCR (dPCR) reference gene panel was successfully developed and validated across synthetic gene fragments, genomic DNA, and cell-free DNA, showing robust linearity, precision, and wide dynamic range.Both the hydrolysis probe and universal (Rainbow™) probe chemistries performed comparably, and the multiplex approach proved superior to single reference gene targets by mitigating bias from genomic instability.

**What are the implications of the main findings?**
The multiplex reference gene panel offers a more reliable method for total DNA quantification, which is crucial for precision medicine applications such as NGS library preparations and copy number variation analysis.This method provides a pathway to establish traceable calibration standards, im-proving quality control and comparability of DNA measurements across laboratories and clinical diagnostics.

**Abstract:**

Precision medicine approaches rely on accurate somatic variant detection, where the DNA input into genomic workflows is a key variable. However, there are no gold standard methods for total DNA quantification. In this study, a pentaplex reference gene panel using digital PCR (dPCR) was developed as a candidate reference method. The multiplex approach was compared between two assay chemistries, applied to healthy donor genomic DNA and plasma cell-free DNA (cfDNA) to measure the ERBB2 (HER2) copy number variation in cancer cell line DNA. The multiplex approach demonstrated robust performance with the two assay chemistries, demonstrating comparable results and a wide dynamic range. Ratios of reference genes were close to the expected 1:1 in healthy samples; however, some small but significant differences (<1.2-fold) were observed in one of the five targets. Expanded relative measurement uncertainty was 12.1–19.8% for healthy gDNA and 9.2–25.2% for cfDNA. The multiplex approach afforded lower measurement uncertainty compared to the use of a single reference for total DNA quantification, which is an advantage for its potential use as a calibration method. It avoided potential biases in the application to CNV quantification of cancer samples, where cancer genome instability may be prominent.

## 1. Introduction

In cancer treatment, precision medicine approaches are often based on therapies which target genetic abnormalities of the tumour. Accurate detection of genetic abnormalities is key when measuring cancer biomarkers for targeted therapeutics and relies on the accurate performance of the genetic and genomic testing methods employed. For example, in breast and bowel cancer, copy number variation (CNV) of Human Epidermal Growth Factor Receptor (*ERBB2*/*HER2*) gene manifests as gene copy number gain, and leads to protein overexpression [1]. This can be targeted by anti-HER2 monoclonal antibody therapies such as trastuzumab. HER2 gene amplification can be measured by fluorescence in situ hybridisation as a relative ratio between the HER2 gene target and a stable reference gene target, typically the CEP17 centromeric region on the same chromosome as HER2 [1]. Digital PCR (dPCR) has been investigated in recent years as an alternative method for *HER2* CNV measurement as it has an advantage of being capable of measuring small fold changes with high precision which is relevant to cell-free DNA (cfDNA) analysis [2]. The genomic landscape of a tumour is vastly heterogeneous, and instability of the genome is one of the distinctive features of cancer [3,4]. This may affect not only cancer biomarkers but also reference genes. For CNV measurement, reference gene targets are often used as a baseline to determine the level of biomarker gene copy number amplification, or deletion. Stable reference genes are important for accurate CNV measurement [5].

Next generation sequencing (NGS), primarily short-read sequencing approaches, in analysis of cancer tissue and, increasingly, blood samples (“liquid biopsies”) are central to the selection of the best treatment for a patient [6]. DNA input into NGS library preparation is an important variable in NGS performance. A variety of methods are used to quantify total DNA concentration, including UV spectrophotometry, fluorescent binding dyes, and quantitative PCR (qPCR) analysis of reference genes. For single copy reference gene loci, these provide an estimate of the number of haploid gene equivalents (GE) in a sample [7]. However there is no gold standard method for quantification of DNA mass or GE concentration [8].

dPCR is a highly sensitive, precise absolute quantification method [9] and has been demonstrated to fulfil the requirements of a reference method for human genetic variants [10,11]. One of the dependencies of dPCR for absolute quantification is the ability to discriminate between partitions containing the targeted amplicons and those without. Partitions with amplicons are identified as positive partitions, and partitions with no amplicons or targets as negative partitions. The determinant signal between positive and negative partitions occurs due to the presence of increased fluorescence, generated using a variety of detection chemistries. Depending on the chemistry used, fluorescence intensity can differ and can influence the peak resolution [12].

Previous studies have highlighted the importance of stable reference gene selection [13] and proposed the use of multiple reference genes as a robust approach for quantification of DNA [5,7]. However, these studies have measured reference genes in uniplex assay format. There are examples of CNV measurement using dPCR in duplex with a single reference gene [5,11,14] which demonstrated more accurate detection of ratios. These studies also highlighted the importance of assessing multiple reference genes, and Vynck et al., 2016 [15] explored in detail the variation in CNV measurement caused by reference genes. In this study, we developed a five-gene multiplex dPCR reference gene panel that can be used for total DNA quantification based on the simultaneous measurement of five reference gene targets as well as a normaliser for measuring the CNV of genes which are amplified or deleted in cancer. During development, five published reference gene assays were selected, and two assay chemistries were explored: hydrolysis “TaqMan” probes and a novel universal probe assay chemistry where sequence-specific probes are not required [16]. Within this study, both assay chemistries were compared in terms of precision and linearity of measurement using the developed reference gene panel across an extensive dynamic range. Measurement uncertainty is a parameter that characterises the dispersion of the measurement result [17] and is a requirement for reference measurement procedures according to ISO 15193 [18]. Measurement uncertainty of reference gene quantification was explored when applying the multiplex approach to different sample types (genomic DNA (gDNA) and cfDNA from blood plasma).

## 2. Materials and Methods

### 2.1. Samples

#### 2.1.1. Human gDNA Restriction Digestion

Prior to dPCR application, 10 units of HindIII restriction endonuclease (NEB, Ipswich, MA, USA) were used for digestion of 1 µg of commercially available human genomic DNA (hgDNA) (Pooled Female, Promega, Madison, WI, USA) at 37 °C for 1 h (Appendix A). Upon digestion, the fragment profile of digested material was confirmed by automated gel electrophoresis (Appendix A). After the restriction digestion, a ten-fold dilution was followed using 1× Tris-EDTA (10 mM TRIS-HCl, 1 mM EDTA, Sigma-Aldrich, St. Louis, MO, USA) and the diluted hgDNA (~688 copies/µL of input concentration based on manufacturer’s value) was used for assay optimisation and validation throughout the study. The diluted template was also used for the preparation of the six dilutions with two-fold serial dilution series using 1× Tris-EDTA as diluent. The diluted material was measured with Qubit Flex fluorometer using double-stranded DNA broad range assay kit (both instrument and kit from ThermoFisher Scientific, Waltham, MA, USA).

Commercially available human genomic DNA (gDNA) isolated from the HCC1954 breast carcinoma cell line with *HER2* copy number gain (ATCC CRL-2338D) and paired healthy control gDNA (ATCC CRL-2339D) isolated from the HCC1954 BL EBV-transformed lymphoblastoid cell line (both ATCC, Manassas, VA, USA) were also digested prior to dPCR application as described above. The digested materials were diluted ten-fold using 1× Tris-EDTA (10 mM TRIS-HCl, 1 mM EDTA) and the diluted gDNA (~552 copies/µL of input concentration of reference gene target based on manufacturer’s value) was used for *HER2* CNV analysis.

#### 2.1.2. gBlocks^TM^ Preparation, Mixing, and Dilution Series

Synthetic double-stranded DNA fragments, gBlock^TM^ (IDT, Coralville, IA, USA), were designed using the whole amplicon size of each reference gene assay (Appendix A). Each of the gBlock^TM^ were prepared according to the manufacturer’s instructions and diluted to ~5000 copies/µL based on dPCR copy number measurement. After the dilution, a mixture of gene fragments was prepared volumetrically with the target ratio of approximately 1:1 between the five reference gene gBlock^TM^ molecules. The six dilutions of gBlock^TM^ mix obtained via two-fold serial dilution series were then prepared using 4 ng/µL Salmon Sperm (Invitrogen, Waltham, MA, USA) as a diluent.

#### 2.1.3. Cell-Free DNA

The Maxwell RSC (Promega, Madison, WI, USA) instrument with the Maxwell^®^ RSC ccfDNA Plasma Kit (Promega, Madison, WI, USA) for large volumes was used for extracting the cfDNA samples from six pooled human plasma (BioIVT, Westbury, NY, USA) from healthy donors. The Maxwell RSC ccfDNA Plasma Large Volume Protocol was followed for the automated cfDNA isolation process using a 2 mL input volume. The elution volume of 75 µL was applied to each extract. The extracts were measured with Qubit Flex fluorometer using double-stranded DNA high sensitivity assay kit (both instrument and kit from ThermoFisher Scientific, Waltham, Massachusetts, USA). Following extraction and initial measurement, the samples were stored at −20 °C.

### 2.2. Digital PCR

#### 2.2.1. Oligonucleotides

A total of five reference genes were selected to assemble the pentaplex (Appendix A) with assays to DCK, HBB, PMM1, RPS27A [19], and RPPH1 [7]. All selected assays were located on different chromosomes and were well characterised assays, already validated in uniplex. University of California Santa Cruz (USCS) Xena Functional Genomics Explorer search for each of the reference genes using The Cancer Genome Atlas Programme’s (TCGA) Colon and Rectal Cancer (COADREAD) study was performed to confirm that there was no systematic genomic instability in the selected reference genes.

For all hydrolysis assays (oligonucleotides from Biosearch Technologies, Lystrup, Denmark), a 20× primer-probe mix was prepared, corresponding to a final reaction concentration of 0.9 µM forward/reverse primers and 0.25 µM probe, whereas for Rainbow^TM^ assays (pxlence, Ghent, Belgium) 20× primer-probe mixes were prepared following the manufacturer’s instructions, corresponding to a final reaction concentration of 0.1 µM Rainbow forward primer, 0.3 µM reverse primer, and 0.125 µM Rainbow probe (Appendix A).

A total of four reference genes were used for the assembly of HER2 multiplex panel pentaplex with hydrolysis probe assays to DCK, PMM1, RPS27A [19], and RPPH1 [7]. These four reference genes were used alongside the HER2 CNV marker [20] for quantification of HER2 copy number ratio. This pentaplex was performed with a 20× primer-probe mix, prepared corresponding to a final reaction concentration of 0.9 µM forward/reverse primers and 0.25 µM probe (Appendix A).

#### 2.2.2. QIAcuity Protocol

Pentaplex assay mixes were prepared with 1× of the corresponding QIAcuity PCR mix (Qiagen, Venlo, The Netherlands) with 1× primer-probe mix and 5.5 µL of template in a prepared volume of 13.2 µL (Appendix A). A reaction volume of 12 µL was transferred into an 8.5 K 96-well nanoplate. Subsequently, nanoplates were loaded onto the QIAcuity Four automated dPCR platform (Qiagen, Venlo, The Netherlands) for priming, cycling, and imaging. Standard built-in priming protocol was selected for both the hydrolysis and Rainbow plates. After the thermal cycling (Appendix A), imaging was performed. Hydrolysis assays were imaged using the default parameters for exposure and gain for Green, Yellow, Red and Crimson detection channels. For the Orange detection channel, an exposure duration of 300 s and gain of 5 was used due to high end-point fluorescence of hydrolysis assays. Rainbow assays were imaged using the default parameters for exposure and gain for all detection channels (Appendix A). No Template Controls (NTCs) were utilised in all experiments, which consisted of Nuclease-free water (*n* = 8, Ambion, ThermoFisher Scientific, Waltham, MA, USA), 1X Tris-EDTA as the hgDNA diluent control (*n* = 4), and Salmon Sperm as the gBlock^TM^ diluent control (*n* = 4). For hydrolysis vs. Rainbow^TM^ comparison, two nanoplates were loaded in parallel and this was repeated three times across three days. Other experiments were performed on a single day. 

### 2.3. Data Acquisition and Analysis

The data were analysed using QIAcuity Software Suite v2.5 (Qiagen, Venlo, The Netherlands). The data were baselined using the negative populations and a common threshold per reference gene per assay chemistry was manually applied to all experiments where possible. Due to suboptimal separation in the high-concentration samples, a separate threshold was applied to the PMM1 reference gene target with Cy5 fluorophore when required for the assay chemistry comparison. Subsequently, the raw data was exported as csv files for further analysis. The sample copy number concentration of target molecules per unit volume was calculated using the average partition volume (Vp) 0.34 nL. All experiments followed the guidelines of the Minimum Information for publication of quantitative digital PCR Experiments (dMIQE) [21]. Further technical information is provided in the Appendix A section and dMIQE table.

### 2.4. Statistical Analysis

Dilution series of gBlocks and hgDNA were analysed by simple linear regression analysis after log-transforming the copy number quantification data with the base 2 using GraphPad Prism v9.5.1 (GraphPad Software, San Diego, CA, USA). The slope and R^2^ values were assessed for the linearity of the samples against the expected values per reference gene target.

Log-transformed data for gDNA dilution series were further analysed using R version 4.2.1 [22] and RStudio version 2025.5.1.513 (RStudio Inc., Boston, MA, USA) with mixed-effects models using the nlme package [23]. To compare differences between reference genes and assay chemistry, the dilution level (sample) and gene were treated as fixed effects and experiment as a random effect, with residual variability (repeatability) stratified according to dilution level. No significant difference was observed between chemistries, so this was omitted from the model. Pairwise differences between reference genes were calculated with 95% confidence intervals (*t* = 4.3, two degrees of freedom). To calculate measurement uncertainties, the model was fitted with gene as a random effect and standard deviations (SD) for gene, experiment, and residual variation (repeatability) were calculated. Measurement uncertainties were calculated by combining relative uncertainties (SD/$\sqrt{n}$) for gene (*n* = 5), experiment (*n* = 1) and repeatability (*n* = 3 × 5 genes). Coverage factor (*k*) was based on the degrees of freedom for gene (4).

Differences between reference genes in cfDNA extracts were analysed using R/RStudio with nlme as above. Data was not log-transformed. Gene was treated as a fixed effect and extract as a random effect. Pairwise difference between reference genes were calculated as for gDNA.

Measurement uncertainties for each cfDNA extract were calculated using SDs for within- and between-reference gene variation from one-way ANOVA tables produced in GraphPad Prism v9.5.1. Measurement uncertainties were calculated by combining uncertainties for gene (*n* = 5) and repeatability (*n* = 3 per gene × 5 genes) as above.

Measurement uncertainties for quantification of total DNA in breast cancer cell line derived gDNA and parental lymphoblast derived gDNA were calculated using SDs for within- and between-reference gene variation from one-way ANOVA tables produced in GraphPad Prism v9.5.1. Measurement uncertainties were calculated by combining uncertainties for gene (*n* = 5) and repeatability (*n* = 2 per gene × 5 genes) as above.

The arithmetic average of reference genes was calculated for gDNA and cfDNA materials for total DNA quantification based on all measurements. HER2 CNV ratio measurements were calculated per dPCR reaction based on single reference genes or all four reference genes.

## 3. Results

### 3.1. Assay Chemistry Performance

The reference gene panel was assembled using five reference gene targets located on separate chromosomes. *HBB*, *RPS27A*, *DCK,* and *PMM1* reference gene targets were previously used for value assignment of certified reference materials in uniplex dPCR application [19]. *RPPH1* assay was also used in a candidate reference method application where *HER2-RPPH1* duplex in dPCR was used for HER2 CNV relative ratio measurement [20]. Previously published hydrolysis probe assays or novel assays using Rainbow chemistry were validated in multiplex format using the QIAcuity dPCR system.

Hydrolysis assays had approximately two-fold higher end-point fluorescence than Rainbow^TM^ assays (Appendix A). For some of the detection channels, the brightness of hydrolysis assays caused a suboptimal condition between neighbouring channels. In Appendix A, a crosstalk pattern between Yellow and Orange detection channels were observed. RPS27A hydrolysis assay detected in the Orange channel had a higher end-point fluorescence which resulted in secondary baseline cluster for RPPH1 assay (visible in Appendix A, grey cluster ~30 RFU with red arrow). This crosstalk pattern had no impact on the quantification of the RPPH1 gene target where the secondary clusters could be excluded by the threshold. Another secondary population observed in HBB where RPS27A end-point fluorescence shifted the HBB-RPS27A double-positive cluster higher than the optimal positioning (visible in Appendix A, blue cluster ~30 RFU with red arrow). This secondary cluster was included in the total positive partition count for HBB gene copy number quantification. It was confirmed that this did not lead to a bias by a comparison with the uniplex format, which showed no significant difference to the multiplex in terms of DNA copy number concentration (Appendix A). Rainbow^TM^ assays had lower end-point fluorescence without crosstalk between detection channels. Overall, optimal peak resolution with minimal rain for both assay chemistries was observed to be able to set thresholds without an impact on the quantification.

### 3.2. Linearity and Chemistry Comparison

To determine the dynamic working range of each reference gene assay per assay chemistry, dilution series of synthetic DNA fragments (“gBlocks^TM^”) were prepared. The quantities measured by each chemistry presented comparable results across a dynamic range over approximately three orders of magnitude. Linear regression analysis demonstrated concordance with the expected copy number concentration values for both assay chemistries. The dilution series exhibited high linearity with R^2^ values between 0.994 and 0.998 (Table 1; linear regression plots and slope table Appendix A and Appendix A). The observed results of gBlock^TM^ mix serial dilution series were tested with paired *t*-test per reference gene for assay chemistry differences and showed no significant difference between assay chemistries with *p*-values > 0.05 (Table 1). The two assay chemistries showed similar precision, with higher variability observed at lower concentrations as expected (Appendix A).

The linearity of each reference gene assay per assay chemistry was also tested with two-fold serial dilution series of pre-digested hgDNA with the input concentration ranging between 600 copies/µL and 20 copies/µL (copies per partition (lambda, λ) provided in Appendix A). Each chemistry demonstrated good linearity against the expected copy number concentration values with slope ranging between 0.989 and 1.029 for both hydrolysis and Rainbow^TM^ (Appendix A). The observed results of the hgDNA serial dilution series were tested with paired *t*-test per reference gene for assay chemistry differences and also showed no significant difference between assay chemistries with *p*-values > 0.05 (Table 1). Likewise, precision was similar across the dilution series for both assay chemistries (Appendix A). Therefore, further statistical analysis was performed with the combined dataset from both chemistries (Appendix A).

### 3.3. Comparison Between Reference Genes

The ratio between reference genes in hgDNA was approximately 1.0 for both assay chemistries, however, some variation (less than 1.2-fold) between targets was observed (Figure 1). Pairwise difference between reference genes in log concentration was calculated with 95% confidence intervals and showed a significant difference between reference gene targets (Appendix A). The HBB copy number concentration was consistently higher (10–20%) than the other four reference gene targets.

### 3.4. Measurement Uncertainty (Analysis of gDNA)

The measurement uncertainty for the estimation of an arithmetic average DNA concentration (genomic copies or GE/µL) based on all five reference genes was calculated taking into account between-gene variation and compared to single gene-based estimates taking into account only individual assay intermediate precision (between-experiment variation and repeatability, Appendix A). Based on a single experiment with triplicate measurements, the standard uncertainty for a single gene was explored and the range of standard measurement uncertainties varied from 5.49 to 13.80% across dilutions (Table 2). In comparison, for the average based on all five reference genes, standard relative measurement uncertainty for hgDNA ranged between 4.34% and 7.14% and expanded measurement uncertainties (95% confidence intervals) ranged from 12.1% to 19.8% (Table 2) for five-gene average. Despite between-gene variation, the precision and measurement uncertainties were improved for the multiplex reference gene panel as compared to single gene application due to the increased number of measurements (15 vs. 3) and reduced residual variation.

The average concentration based on all five reference genes for hgDNA sample D1 was converted to mass concentration and compared to fluorometry measurements. Fluorometry-based mass concentration measurements were obtained from eight replicate measurements of hgDNA D1 by Qubit Flex fluorometer. The mean value was obtained from dPCR 1.99 ng/µL for hydrolysis and 2.00 ng/µL for Rainbow^TM^ assay chemistry. The total mass concentration observed from fluorometric measurements was 1.62 ng/µL (1.3% CV), which was 1.2-fold lower than dPCR values. Both dPCR and fluorometric measurements were compared to the manufacturer’s value, 2.49 ng/µL (total dilution factor of 100 applied to the stock concentration of 249 ng/µL), which was based on UV spectrophotometer-based measurement. There were 1.24-fold and 1.53-fold differences observed for dPCR and fluorometer values, respectively, when compared to the manufacturer’s measurement value (Appendix A).

### 3.5. Analysis of cfDNA Extracted from Plasma

The reference gene panel using hydrolysis assay chemistry was applied to six samples of cfDNA extracted from pooled healthy donor plasma (Figure 2A). The dPCR input concentration ranged between 53 copies/µL and 65 copies/µL (copies per partition (lambda, ʎ) provided in Appendix A), which fell within the validated dynamic range of the reference gene panel. The observed concentration of RPPH1 was ~21% lower than the other four reference gene targets (Figure 2B). Measurement uncertainty was calculated based on the single experiment with triplicate measurements per reference gene and relative standard uncertainty ranged from 3.3 to 9.1%, with expanded uncertainties (95% confidence interval) between 9.2 and 25.2% (*k* = 2.78) (Figure 2A, Appendix A).

The average concentration based on all five reference genes was calculated per mL plasma and converted to mass concentration (ng/mL plasma) to compare with fluorometry measurements. Fluorometry-based mass concentration measurements were obtained from three replicate measurements of all cfDNA extracts by Qubit Flex fluorometer. The mean value of all cfDNA extracts obtained from dPCR was 7.50 ng/mL (16.05% CV), ranging between 6.57 ng/mL (7.31% CV) and 8.36 ng/mL (20.31% CV). The total mass concentration observed from fluorometric measurements was 4.60 ng/mL (6.75% CV), ranging between 4.25 ng/mL (2.55% CV) and 5.05 ng/mL (1.71% CV). The overall average of dPCR measurements was ~1.6-fold higher than the average fluorometric measurements for all cfDNA extracts (Appendix A).

### 3.6. Analysis of Genomic DNA Material from Breast Carcinoma Cell Line

The reference gene panel was also applied to CNV analysis using a *HER2* multiplex panel where the *HBB* reference gene target was replaced with the *HER2* CNV marker. Both the five-reference gene and HER2 multiplexed panels were applied to HER2 amplified (HER2+ gDNA) and parental (lymphoblast gDNA) gDNA materials to estimate the total DNA and HER2 copy number gain, respectively. The most variation between reference genes was observed in HER2+ gDNA, where expanded measurement uncertainty was 34.75%. The HBB gene target measured the lowest (~400 copies/µL of DNA input), followed by the RPS27A and PMM1 gene targets (~550 copies/µL of DNA input). The RPPH1 and DCK gene targets measured at a higher level (~700–900 copies/µL of DNA input). The expanded measurement uncertainty for parental lymphoblast (wild type) gDNA was 9.64% (Figure 3A), similar to that observed for hgDNA D1 (Table 2).

The variation in reference genes affected the HER2 relative ratio measurement in HER2+ gDNA materials (Figure 3B). The differences in HER2 ratio obtained from single reference genes between the lowest and highest was ~1.8 fold. The HER2 ratio based on all four reference genes was close to the average HER2 ratio of ~50 copies per haploid genome.

## 4. Discussion

dPCR applications utilising assay multiplexing have gained momentum within the scientific community in recent years, capitalising on technological advancements offered by instrument manufacturers combined with those in fluorescent oligonucleotide chemistries. Standard hydrolysis probes have been central to dPCR applications due to their high specificity, high peak resolution, and multiplexing compatibility [24]. However, the requirement for a sequence-specific probe can be disadvantageous due to the knowledge requirement for complex assay design, the compatibility with various sample types that have differential fragment size profiles, and the initial assay optimisation costs. When considering these, universal probe chemistries may offer a solution with reduced assay design complexity and initial costs [24]. An additional potential benefit of universal probes is their applicability to samples with fragmented profiles, such as cfDNA. The primer pairs can be designed with smaller amplicon sizes without the need for a sequence-specific probe. Within this study, a comparison between standard hydrolysis and novel universal Rainbow^TM^ [16] probe chemistries was performed using a pentaplex dPCR multiplexing strategy and the assay performance of the multiplexed formats was established. Both the hydrolysis and Rainbow^TM^ assay chemistries demonstrated comparable linearity, dynamic range, and measurement precision, achieving a relative standard deviation (RSD) of <25% at the lowest dilution points of gDNA (equivalent to ~20 copies/µL input concentration). Within this study, QIAcuity nanoplates with 8.5 K partitions were used; however, greater precision may be obtained for lower copy number concentration samples such as cfDNA by the use of dPCR nanoplates with higher numbers of partitions.

Variability in reference gene measurement was another aspect evaluated within this study. Healthy female donor-derived hgDNA was used for the optimisation and validation of the multiplexed reference gene panel. Generally, the ratio between the five targets was close to the expected 1:1 ratio; however, some small but significant differences between the five reference genes were observed. In pooled hgDNA, the *HBB* reference gene target was consistently higher than other reference genes by ≤1.2-fold. Variations in *HBB* gene amplification have been associated with various genetic disorders including β-thalassemia [25], suggesting that caution might need to be exercised when considering *HBB* as reference gene for total DNA quantification. This further highlights the importance of making an informed choice about reference gene selection, as it is a potential contributor to bias in quantitative measurements.

Assessing the total DNA concentration from complex sample matrices such as cfDNA in plasma can be challenging due to the associated short (~160 bp) fragment profile and low sample concentration [26]. When estimating the total DNA concentration of cfDNA by PCR-based methods, amplicon size plays an important role. Given that the fragment size of cfDNA may be lower in cancer patients compared to healthy individuals [27,28], shorter amplicon sizes may be considered advantageous. The other key element is the assay sensitivity due to the lower concentration ranges of cfDNA [26]. The applicability of the reference gene panel to cfDNA was assessed by an extended range of serial dilution series to determine the working dynamic range. The hgDNA serial dilution series from ~600 to ~20 copies/µL demonstrated the expected reduction in the intermediate precision and increase in the relative uncertainty with decreasing template concentration. The reference gene panel was used for total DNA concentration estimations of cfDNA obtained from pooled healthy donor plasma. The reference gene panel cfDNA measurement showed a significant difference between *RPPH1* and the other four reference gene targets, where *RPPH1* measured lower than the other reference genes. Surprisingly, RPPH1 had the smallest amplicon size (64 bp) within all five reference genes. This variation might be linked with the heterogeneous fragment profile of cfDNA causing differences among reference genes, for example, due to nucleosomal phasing [29]. This highlights the impact of varying sample types on quantification of alternative reference genes. The current results highlight that potential biological sources of variation can introduce bias into total DNA quantification if only a single reference gene is measured, whereas the multiple reference gene estimate provides a more conservative approach.

Measurement of more than a single reference gene in cancer samples may be advantageous in case a single locus is affected by genomic instability. Utilising five reference genes is potentially advantageous to achieve a robust method for measuring total DNA quantification for an NGS workflow as well as acting as a normaliser for CNV analysis. This measurement can be expressed as either mean copy number concentration or haploid genome equivalents (GE) where it is assumed that the targeted reference gene is present at a single copy per haploid genome. Within this study, HER2 CNV analysis was performed by calculating the number of HER2 copies per haploid genome in breast carcinoma cell line gDNA and its parental cell line gDNA. This was compared between either HER2 relative ratio per single reference gene or HER2 relative ratio from the average of four reference genes. The HER2 relative ratio calculations per single reference gene approach demonstrated high variability, whereas the CNV ratio based on all four genes reduced the impact of reference gene variability. Reference gene stability can further be evaluated in the future in primary samples from cancer patients.

The performance of multiplexed dPCR is reliant on multiple factors such as the quantity, quality, and complexity of the template [30]. Within this study, we utilised three different template types to evaluate the developed pentaplex assay for total DNA measurement. Higher quantity input range was tested with lower complexity gBlock^TM^ mix, where smaller double-stranded molecules were fully accessible for target amplification. Using higher template quantities can be advantageous for single target dPCR measurements which can provide more balanced proportions of negative and positive number of partitions with a lower %CV for partitioning. However, with multiplexing dPCR, multiple occupancy of targets can affect the precision of the dPCR assay performance for total DNA measurement. Therefore, a lower input range was evaluated with digested hgDNA, where non-linked DNA molecules were fully accessible for target amplification at a limiting dilution range. hgDNA is prone to mechanical damage due to handling and storage. This may lead to changes to target sites which can affect the total DNA copy number concentration.

During the development of the reference gene panel, two main possible limitations are considered where the developed method might introduce bias to the measurement. The first limitation is the inability to distinguish DNA strandedness, where single-stranded DNA molecules can cause over-quantification [8,31]. dPCR measurements of total DNA in hgDNA and cfDNA samples demonstrated distinct differences to mass concentration measurements obtained from fluorometry, which targets double-stranded DNA molecules (Appendix A); however, the standards used for fluorometric assays are not metrologically traceable. The difference observed between dPCR- and fluorometry-based total DNA quantification might be associated with the impact of single-stranded DNA amplification in dPCR. For an accurate total DNA quantification and characterisation, mass spectrometry-based methods such as isotope-dilution mass spectrometry (IDMS) have been recommended due to its full traceability to the International System of Units (SI) [32,33].

Secondly, in cancer genomes where genomic polyploidy occurs heavily, reference gene-based copy number estimation might not always reflect haploid genome quantification per cell. These factors can be determined with the use of orthogonal methods such as sequencing along with well-characterised calibration materials [31]. Another aspect that might require investigation is further validation of the developed method across different dPCR platforms to capture instrument- and partition volume-related bias introduced to the measurement [10]. Despite these limitations, the developed method demonstrated a robust performance with a wide dynamic range. This can be beneficial when utilising lower concentration samples for total DNA quantification, as demonstrated in our cfDNA application. Furthermore, the reference gene panel successfully demonstrated the ability to capture heterogeneity across various samples through multiple reference gene targets. This is a key applicability of the method which demonstrates the importance of utilising multiple reference genes for CNV measurements.

Quality control metrics hold a key role throughout the NGS workflow. Numerous factors have been shown to bias NGS results, especially those attributed to sample type, quantity, and quality [34]. Accurate measurement of the sample input and sequencing libraries are necessary throughout the NGS workflow to ensure traceable, high quality data outputs. A traceable measurement system for total DNA concentration would enable potential sources of measurement error affecting the performance of the method to be pinpointed. The developed multiplex dPCR method can provide improved confidence in the total DNA measurement by the provision of an average value with its measurement uncertainty. It can be used to calibrate standards for commonly used quantification methods to achieve a standardised and traceable quality assessment. Geometric average of reference gene measurement can be considered for total DNA quantification [35]; however, unlike gene expression studies, measurement of nuclear DNA by reference genes are unlikely to vary over a log scale. Commonly used relative quantification methods for total DNA measurement, such as fluorescent binding dye assays, electrophoretic platforms, and qPCR require calibration, may benefit from higher order traceable standards (reference materials and reference methods) to ensure the stability of their performance. Higher order reference materials such as NIST SRM 2372a [19], in combination with dPCR reference methods, could fulfil this function of assigning values to end-user calibration and QC materials. Using dPCR to apply reference gene panel multiplexing strategies could also be deployed to in-house materials to generate quality control standards assigned in GE concentration where the GE concentration may be converted to mass (ng) concentration to achieve comparative measurement units [36].

## 5. Conclusions

In this study, a novel multiplexed reference gene approach to measure total DNA was validated using various sample types including synthetic gene fragments, hgDNA, cfDNA, and breast cancer cell line gDNA. A parallel comparison of standard assay technology with the novel universal assay technology, Rainbow^TM^, was performed using synthetic gene fragments and hgDNA, which demonstrated the robustness of the approach with equivalent performance in terms of linearity, dynamic range, and precision. Variability between the five reference genes was observed when using different sample types; therefore, the reference gene selection needs to be validated for specific sample types. The developed method can provide a route for calibration of GE concentration and conversion to mass concentration to support quality assurance of genomic workflows. Genomic instability in cancer can lead to variation in reference gene targets, which can be statistically investigated for outliers when estimating the average reference gene measurement. The developed method demonstrated the advantages of using multiple reference genes when measuring CNV markers. The developed method can be further explored to estimate the total DNA measurement for NGS workflows along with higher order traceable standards to provide improved confidence in genomic analysis.

## Figures and Tables

**Figure 1 cells-14-01544-f001:**
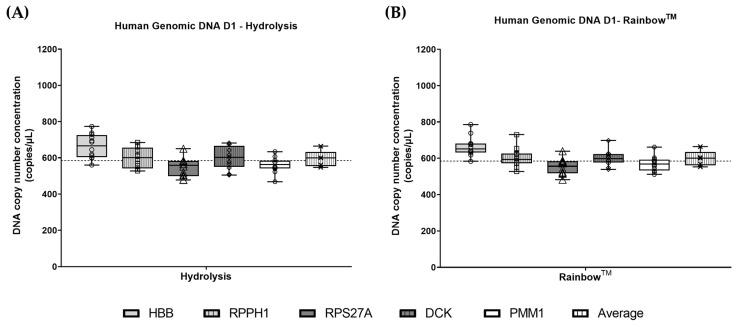
Reference gene variability measurements in hgDNA samples: (**A**) Reference gene comparison using hgDNA for hydrolysis assay chemistry; (**B**) Reference gene comparison using hgDNA for Rainbow^TM^ chemistry. Dashed lines are overall median. Boxplots represent the minimum to maximum (whiskers) with interquartile range (box), with all data points shown as symbols.

**Figure 2 cells-14-01544-f002:**
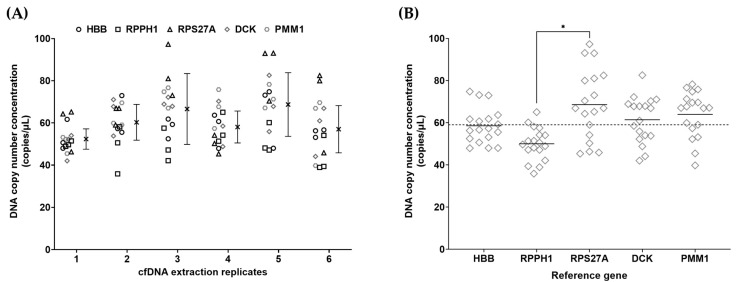
Reference gene target measurement of pooled donor plasma derived cell-free DNA (copies/µL eluate) shown according to sample (**A**) and reference gene (**B**). Data points reflect individual measurements. (**A**) Error bars reflect expanded measurement uncertainty with cross represents the mean value. (**B**) Lines are mean for each reference gene; dashed line shows overall mean. * *p* < 0.05.

**Figure 3 cells-14-01544-f003:**
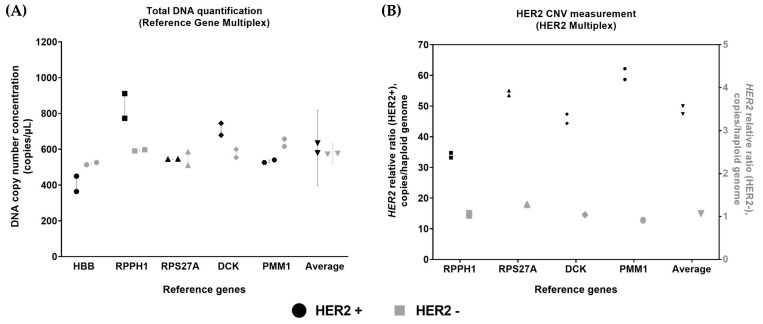
Reference gene measurement and HER2 ratio analysis of breast carcinoma cell line-derived hgDNA. (**A**) Reference gene panel was tested using HER2+ gDNA and its matched lymphoblast gDNA (HER2− gDNA). Error bar for the individual genes show range. The error bar for the average shows the expanded measurement uncertainty.; (**B**) HER2 multiplex panel was tested using HER2+ gDNA and its matched lymphoblast gDNA (HER2−). Symbols in black represent HER2+ gDNA, and symbols in grey represent HER2− gDNA. Circle, square, triangle, diamond, hexagon, and down-pointing triangle represent *HBB, RPPH1, RPS27A, DCK, PMM1* and the average respectively. Each symbol shows individual measurements.

**Table 1 cells-14-01544-t001:** gBlock^TM^ Mix and hgDNA linear regression analysis and paired *t*-test results.

	gBlock^™^ Mix			hgDNA		
Reference Genes	Hydrolysis R^2^	Rainbow^™^ R^2^	*p*-Values	Hydrolysis R^2^	Rainbow^™^ R^2^	*p*-Values
HBB	0.996	0.994	0.546	0.988	0.985	0.612
RPPH1	0.997	0.996	0.949	0.983	0.988	0.134
RPS27A	0.997	0.996	0.752	0.976	0.985	0.997
DCK	0.998	0.996	0.466	0.989	0.986	0.234
PMM1	0.996	0.996	0.195	0.984	0.987	0.809

**Table 2 cells-14-01544-t002:** The measurement uncertainty of single vs. five reference gene-based estimates for a two-fold serial dilution of hgDNA.

Dilution	Five Gene Average, GE/µL	Relative Standard Uncertainty (%)— Single Gene	Relative Standard Uncertainty (%)— Five Gene	Relative Expanded Uncertainty (%)— Five Gene *
D1	664	5.49	4.34	12.1
D2	360	6.40	4.59	12.7
D3	179	6.23	4.54	12.6
D4	83	6.30	4.56	12.7
D5	41	10.49	5.90	16.4
D6	21	13.80	7.14	19.8

* Coverage factor (k) = 2.78 (95% confidence interval).

## Data Availability

All raw data will be made available upon request.

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
