# Peer review of "Multiplexed Digital PCR Reference Gene Measurement for Genomic and Cell-Free DNA Analysis"

_cells, 2025, doi:10.3390/cells14191544_

Round 1
Reviewer 1 Report
Comments and Suggestions for Authors
The authors test a multiplex of five previously developed reference gene assays using digital PCR. They test the multiplex with two different chemistries and on three different materials (hgDNA pool, gBlocks, and cfDNA pools). The multiplex dPCR is described as a method to more precisely quantify DNA before NGS analysis or a method to normalize data for copy number variation.
The multiplex is described in detail, and the methodological presentation is overall satisfactory, but the proposed advantages of the method are not actually demonstrated. The presented data supports that the method is robust (although with differences between sample materials), but it would greatly improve the manuscript, if it was somehow demonstrated, that NGS performance is improved if the multiplex is used for quantification of samples, or that normalization of CNV analyses is more accurate with these specific assays. How can this be demonstrated except by using individual patient samples and test performance in NGS or CNV analyses?
HER2 is mentioned in the introduction as a clinically relevant biomarker for CNV analyses, but it is stated that this marker is measured by FISH. What is the relevance for the presented method? Are there other examples/references where dPCR is used?
It should be described how the five assays included in the multiplex were selected and whether they have proven to be suitable reference genes (i.e. stable in cancer).
Four assays from the NIST publication that describes 10 assays are included – why were these four chosen?
It should be checked if all supplementary figures and tables are mentioned in the manuscript and numbered according to when they are mentioned in the manuscript. I did not find any reference to supplementary figure S1, S2, S3 and S4 or supplementary table 11.
The plasma pools are not mentioned under specimen in the dMIQE checklist.
Figure S8 is mentioned in the dMIQE checklist as illustration for fragment size confirmation. Is Figure S9 the correct figure?
Extraction blanks are stated as test for inhibitors. How?
It is not clear what material is tested in figure 1. Are dPCR results similar for all sample types?
Was any comparison of single- vs. multiplex performed? Was any optimization of the multiplex done? Primer/probe concentrations, temperature, etc. The assays have been published as single assays with different protocols.
The paragraph ‘3.1 Assay chemistry performance’ is difficult to read (e.g. ‘This secondary cluster contributed to the total HBB gene copy number quantification therefore it did not affect the quantification of the HBB’). Please rephrase.
How should results from the multiplex be used in order to obtain more precise quantification of DNA? As an average value? What if one of the values is an outlier?
Line 387-391 is unclear. Please clarify.
Author Response
Please see attached document for responses to Reviewer 1

Reviewer 2 Report
Comments and Suggestions for Authors
The manuscript can be accepted after minor revisions. Please add the following information:
Methods section:
- The manufacturers (name, country) of the reagents and platforms used.
- For cfDNA testing, plasma samples from healthy donors were used.
- 'Corresponding QIAcuity PCR mix' – please specify.
Add a paragraph about limitations:
- Were 8.5K plates used for both gDNA and cfDNA measurements? For the latter, 26K plates are recommended.
- No other dPCR platforms were tested.
- No cancer cfDNA samples were tested.
- Please add anything else that is relevant.
Author Response
Please see attached document for responses to Reviewer 2

Reviewer 3 Report
Comments and Suggestions for Authors
Dear Authors,
the manuscript presents a well-designed study; however, some improvements are needed, according to the following suggestions:
- In 2.2. Digital PCR 108 , 2.2.1. Oligonucleotides, line 115 indicate the Manufacturer.
- In 3.5 Analysis of cfDNA extracted from plasma give an explication for the following finding: The observed concentration of RPPH1 was ~21% lower than the 263 other four reference gene targets (Figure 3B).
- In 3.2 Linearity and chemistry comparison clarify the software used for statisticals
- In 3.4 Measurement uncertainty (analysis of gDNA) and3. 5 Analysis of cfDNA extracted Indicate the software used for the calculation of Measurement uncertainty.
- In discussion indicate the limitation of the developed methods
- Expand conclusions also addressing the limitations of the study.
- Add a graphical overview of the workflow.
- Improve resolution of Fig. 1 and include it as supplementary material.
- Improve resolution of Fig.3A and 3B
- Revise the text because it appears dense in some places and could benefit from clearer, shorter sentences.
Author Response
Please see attached documents for responses to Reviewer 3

Round 2
Reviewer 1 Report
Comments and Suggestions for Authors
Thank you for the revised version of the manuscript. The authors have addressed the issues that were pointed out. It has greatly improved the manuscript that additional data has been added and the context is more clear with the revised introduction and discussion.